# Domain–domain interactions determine the gating, permeation, pharmacology, and subunit modulation of the IKs ion channel

**Mark A Zaydman[1], Marina A Kasimova[2,3], Kelli McFarland[1], Zachary Beller[1], Panpan Hou[1], Holly E Kinser[1], Hongwu Liang[1], Guohui Zhang[1], Jingyi Shi[1], Mounir Tarek[2,4], Jianmin Cui[1]\***

[1]Department of Biomedical Engineering, Center for the Investigation of Membrane Excitability Diseases, Washington University in St Louis, St Louis, United States; [2]Theory, Modeling, and Simulations, UMR 7565, Université de Lorraine, Nancy, France; [3]Lomonosov Moscow State University, Moscow, Russia; [4]UMR 7565, Centre National de la Recherche Scientifique, Vandoeuvre-lés-Nancy, France

**Abstract** Voltage-gated ion channels generate electrical currents that control muscle contraction, encode neuronal information, and trigger hormonal release. Tissue-specific expression of accessory (β) subunits causes these channels to generate currents with distinct properties. In the heart, KCNQ1 voltage-gated potassium channels coassemble with KCNE1 β-subunits to generate the $I_{Ks}$ current (*Barhanin et al., 1996*; *Sanguinetti et al., 1996*), an important current for maintenance of stable heart rhythms. KCNE1 significantly modulates the gating, permeation, and pharmacology of KCNQ1 (*Wrobel et al., 2012*; *Sun et al., 2012*; *Abbott, 2014*). These changes are essential for the physiological role of $I_{Ks}$ (*Silva and Rudy, 2005*); however, after 18 years of study, no coherent mechanism explaining how KCNE1 affects KCNQ1 has emerged. Here we provide evidence of such a mechanism, whereby, KCNE1 alters the state-dependent interactions that functionally couple the voltage-sensing domains (VSDs) to the pore.

**\*For correspondence:** jcui@wustl.edu

**Competing interests:** The authors declare that no competing interests exist.

**Reviewing editor**: Richard Aldrich, The University of Texas at Austin, United States

## Introduction

Voltage-gated ion channels sense changes in membrane voltage and respond by opening or closing a pore through which selected ions cross the membrane, generating a transmembrane current. These channels consist of four voltage-sensing domains (VSDs) surrounding a central pore. In voltage-gated potassium (Kv) channels, this structure results from the tetrameric assembly of Kv-α subunits, each of which contain six transmembrane-spanning segments (S1–S6). S1–S4 of each subunit forms a voltage-sensing domain (VSD), and S5–S6's from all four subunits form the pore. Sensing of membrane voltage occurs within the VSDs, which contain a mobile S4 segment with several highly conserved basic residues. At depolarized voltages, the forces of the membrane electric field on these positively charged residues promotes the outward displacement of S4 toward its activated state (*Papazian et al., 1995*; *Larsson et al., 1996*; *Silva et al., 2009*; *Wu et al., 2010a*; *Delemotte et al., 2011*; *Jensen et al., 2012*). The pore contains the ion permeation pathway that can be opened and closed by the reorientation of the intracellular portions of the S6 helices (*Liu et al., 1997*; *Webster et al., 2004*; *del Camino and Yellen, 2001*; *Jiang et al., 2002*). Critical to voltage-dependent gating are the interactions between the VSDs and the pore, which couple the activation of the VSD to the opening of the pore, resulting in a voltage-gated conductance (*Chen et al., 2001*; *Lu et al., 2001*, *2002*; *Long et al., 2005*; *Lee et al., 2009*; *Zaydman et al., 2013*).

**eLife digest** Cells are surrounded by a membrane that prevents charged molecules from flowing directly into or out of the cell. Instead ions move through channel proteins within the cell membrane. Most ion channel proteins are selective and only allow one or a few types of ion to cross. Ion channels can also be 'gated', and have a central pore that can open or close to allow or stop the flow of selected ions. This gating can be affected by the channel sensing changes in conditions, such as changes in the voltage across the cell membrane.

Research conducted more than half a century ago—before the discovery of channel proteins—led to a mathematical model of the flow of potassium ions across a membrane in response to changes in voltage. This model made a number of assumptions, many of which are still widely accepted. However, Zaydman et al. have now called into question some of the assumptions of this model.

Based on the original model, it has been long assumed that the voltage-sensing domains that open or close the central pore in response to changes in voltage must be fully activated to allow the channel to open. It had also been assumed that the voltage-sensing domains do not affect the flow of ions once the channel is open. Zaydman et al. have now shown that these assumptions are not valid for a specific voltage-gated potassium channel called KCNQ1. Instead, this ion channel opens when its voltage-sensing domains are either partially or fully activated. Zaydman found that the intermediate-open and activated-open states had different preferences for passing various types of ion; therefore, the gating of the channel and the flow of ions through the open channel are both dependent on the state of the voltage-sensing domains. This is in direct contrast to what had previously been assumed.

The original model cannot reproduce the gating of KCNQ1, nor can any other established model. Therefore, Zaydman et al. devised a new model to understand how the interactions between different states of the voltage-sensing domains and the pore lead to gating. Zaydman et al. then used their model to address how another protein called KCNE1 is able to alter properties of the KCNQ1 channel.

KCNE1 is a protein that is expressed in the heart muscle cell and mutations affecting KCNQ1 or KCNE1 have been associated with potentially fatal heart conditions. Based on the assumptions of the original model, it had been difficult to understand how KCNE1 was able to affect different properties of the KCNQ1 channel. Thus, for nearly 20 years it has been debated whether KCNE1 primarily affects the activation of the voltage-sensing domains or the opening of the pore. Zaydman et al. found instead that KCNE1 alters the interactions between the voltage-sensing domains and the pore, which prevented the intermediate-open state and modified the properties of the activated-open state. This mechanism provides one of the most complete explanations for the action of the KCNE1 protein.

In their pioneering work on the action potential of the squid giant axon, Hodgkin and Huxley empirically derived a model for the K$^+$ conductance in which a transmembrane pathway is gated by four voltage-dependent particles (*Hodgkin and Huxley, 1952*). The legacy of the Hodgkin and Huxley model can still be found in the current Kv channel models, which assume that (1) VSD activation and pore opening are two-state (i.e., all or none) processes and (2) gating and permeation are independent so that the VSD conformation changes the probability of pore opening, but does not affect the properties of the open pore. Several recent studies call into question the assumption that VSD activation and pore opening are two-state processes. Computational and experimental studies have demonstrated that VSD activation actually occurs in a series of stepwise transitions due to salt bridge interactions between the basic residues on S4 and acidic residues on S1 and S2, which define resting, intermediate, and activated states (*Papazian et al., 1995*; *Tiwari-Woodruff et al., 1997*; *Wu et al., 2010a*; *Delemotte et al., 2011*; *Jensen et al., 2012*; *Lacroix et al., 2012*). With regards to pore opening, recordings of single channel currents from Kv channels revealed multiple open states discernable by their different conductance levels (*Chapman et al., 1997*), although the identities of these subconductance states remain unclear. In our present study of KCNQ1 (Kv7.1, KvLQT1) channels, we found that both the intermediate and fully-activated states of the VSD yielded robust pore-opening.

Remarkably, the intermediate-open and activated-open states had different permeation and pharmacological properties revealing that VSD-pore interactions determine both open probability and open conformation, demonstrating that gating and permeation are not independent.

KCNQ1 channels generate currents with very different properties as a result of tissue specific expression of KCNE family accessory subunits (*Abbott, 2014*). In the heart, channels formed by KCNQ1 and KCNE1 subunits are responsible for the slow-delayed rectifier potassium current, $I_{Ks}$ (*Barhanin et al., 1996*; *Sanguinetti et al., 1996*), which plays a critical role in limiting action potential duration when beta-adrenergic tone is elevated. The importance of this response is highlighted by a large set of loss-of-function mutations of KCNQ1 or KCNE1 that have been associated with Long QT Syndrome and result in an elevated risk of fatal arrhythmias during times of stress (*Paavonen et al., 2001*; *Schwartz et al., 2001*; *Hedley et al., 2009*). Although KCNE1 is a small, single-membrane-spanning peptide, its coassembly dramatically alters every physiologically relevant property of the KCNQ1 channel: voltage-dependence, current kinetics, inactivation, current amplitude, single channel conductance, selectivity, and pharmacology (*Wrobel et al., 2012*; *Sun et al., 2012*). The mechanism of how KCNE1 modulates KCNQ1 has been a longstanding topic of debate with several groups arguing that KCNE1 alters VSD activation (*Nakajo and Kubo, 2007*; *Ruscic et al., 2013*), several other groups arguing that KCNE1 alters pore opening (*Rocheleau and Kobertz, 2008*; *Osteen et al., 2010*), and several reports claiming that KCNE1 directly contributes to the inner structure of the pore (*Wang et al., 1996*; *Tai and Goldstein, 1998*). However, these mechanisms are not able to simultaneously account for the effects of KCNE1 on gating and the observed changes in permeation and pharmacology.

Here we made three observations regarding the function of homomeric KCNQ1 channels that were not previously reported. First, we found that VSD activation occurs in two resolvable steps through a stable intermediate state. Second, we observed that the intermediate-state of the VSD is sufficient to promote KCNQ1 channel opening, resulting in both intermediate-open and activated-open states. Third, we observed that the intermediate-open and activated-open states have different permeation and pharmacological properties. With these critical observations, we were able to reexamine how KCNE1 affects KCNQ1. We found that coexpression of KCNE1 prevented the intermediate-open state and changed the properties of the activated-open state. The apparent decoupling of pore opening from the resting to intermediate transition of the VSD suggested that KCNE1 changes how the VSD and pore interact. Consistent with this hypothesis, changing the VSD-pore interactions directly via a single point mutation also prevented the intermediate-open state and modified the properties of the activated-open state. Using a kinetic model, we demonstrated that the effects of KCNE1 on VSD-pore interactions, as suggested by our data, are sufficient to simultaneously explain most of the changes in activation gating without any direct impacts on VSD activation or pore opening. Furthermore, as VSD-pore interactions were found to determine the open-pore properties, the effects of KCNE1 on permeation and pharmacology could also be rationalized. Therefore we conclude that altering VSD-pore interactions is likely the primary mechanism through which KCNE1 modulates KCNQ1.

## Results

Voltage-clamp fluorometry (VCF) simultaneously monitors VSD-activation and pore opening (*Mannuzzu et al., 1996*). In our VCF records, a fluorophore attached to the S3–S4 linker of pseudo-WT KCNQ1 channels (with mutations C214A/G219C/C331A) reports on conformational changes associated with VSD activation (*Figure 1A–H*, green), while the ionic currents report the opening of the pore (*Figure 1A–H*, black) (*Osteen et al., 2010*; *Zaydman et al., 2013*). We observed multiple components of the fluorescence signals for KCNQ1 (*Figure 1A–D*) and, as recently reported (*Barro-Soria et al., 2014*), for KCNQ1+KCNE1 (*Figure 1E–H*). Most of the total change in fluorescence intensity was due to a fast component occurring at hyperpolarized voltages ($F_{main}$), but a small additional change was observed due to a slow component occurring at highly depolarized voltages ($F_{high}$). $F_{high}$ was more prominently observed when KCNQ1 channels were labeled with a different dye (*Figure 1—figure supplement 1A,B*), or coexpressed with a mutant KCNE1 (*Figure 1—figure supplement 1C,D*).

Strikingly, KCNE1 shifted the conductance-voltage (GV) relationship so that it correlated with a different component of the fluorescence-voltage (FV) relationship. In the absence of KCNE1, pore opening (i.e., GV) occurred in a similar voltage range as $F_{main}$ (*Figure 1A*, *Figure 1—figure supplement 1B*). In contrast, in the presence of KCNE1, pore opening was not observed unless more depolarized voltages were applied, as with $F_{high}$ (*Figure 1E*, *Figure 1—figure supplement 1D*). These FV-GV

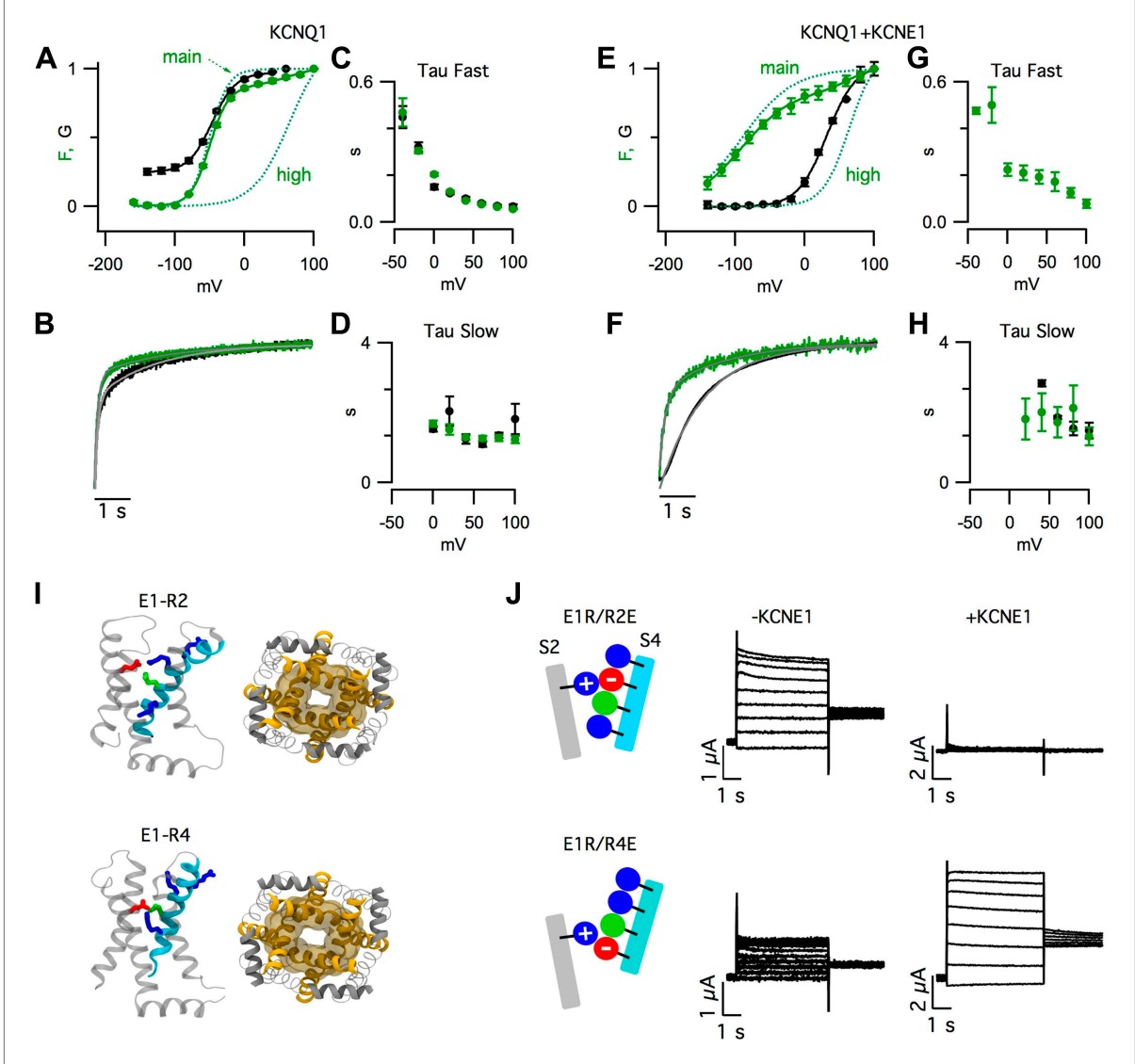

**Figure 1**. KCNE1 suppresses the intermediate-open state of KCNQ1. (**A–H**) Fluorescence (green) and current (black) signals from *Xenopus oocytes* injected with cRNA encoding pseudo-WT (C214A/G219C/C331A) KCNQ1 alone (KCNQ1, **A–D**) or coinjected with cRNAs encoding pseudo-WT KCNQ1 and KCNE1 (KCNQ1+KCNE1, **E–H**). The cells were labeled with Alexa 488 C5-maleimide. (**A** and **E**) GV and FV relationships (solid) with the main and high voltage FV components plotted (dotted lines). (**B** and **F**) normalized fluorescence and current responses to a 60 mV pulse shown with fits (thin grey lines) to a single- or bi-exponential function. Averaged fast (**C** and **G**) and slow (**D** and **H**) tau values of fluorescence and current responses to various voltage pulses. (**I**) Intermediate- (E1-R2, top) and activated- (E1-R4, bottom) state homology models of KCNQ1 after 100 ns of MD simulation. Side view of one VSD (left) and bottom view of the pore (right). (**J**) Currents from the cells expressing E160R/R231E (E1R/R2E, top) or E160R/R237E (E1R/R4E, bottom) both alone (−KCNE1, middle) or with KCNE1 (+KCNE1, right).

The following figure supplements are available for figure 1:

**Figure supplement 1**. Improved resolution of Fhigh.

**Figure supplement 2**. GV/FV relationships are maintained in channels the mutation R243Q in KCNQ1.

**Figure supplement 3**. MD simulations predict that, unlike the resting-state, both the intermediate- and activated-states of the VSD stabilize pore opening through state-dependent protein and lipid interactions.

*Figure 1. Continued on next page*

*Figure 1. Continued*

**Figure supplement 4**. VSD mutations reveal that KCNE1 suppresses currents from intermediate-open states and increases those from activated-open states.

**Figure supplement 5**. Surface membrane expression of E1R/R2E and E1R/R4E.

correlations were not likely to be coincidental because they were maintained in the presence of a KCNQ1 mutation, R243Q, that shifted the voltage dependence of channel opening but did not change the correlation of the GV to Fmain or Fhigh in the absence or precense of RKK/EEE KCNE1, respectively (*Figure 1—figure supplement 2*).

KCNE1 also altered the time-dependence of pore opening. KCNQ1 current onset had two exponential components following the two timecourses of the fluorescence (*Figure 1B–D*). With KCNE1, the channels remained closed during the fast fluorescence increase and, after this initial delay, the channels opened with a single timecourse, similar to the slow fluorescence component (*Figure 1F–H*). Both the steady-state and kinetic VCF data can be easily explained if one VSD transition ($F_{main}$) is sufficient to open KCNQ1, but an additional transition ($F_{high}$) is required for KCNQ1+KCNE1.

The observation of two fluorescence components (*Figure 1A–H*) is consistent with the suggestion from previous studies (*Silva et al., 2009*; *Wu et al., 2010a*) that, in KCNQ1, VSD activation occurs in two sequential transitions due to electrostatic interactions between E160 (E1) in S2 and arginine residues in S4. These interactions stabilize discrete resting, intermediate, and activated states. We built homology models of KCNQ1 with the VSDs in states where E1 forms a salt bridge with R228 (R1), R231 (R2), or R237 (R4) (*Figure 1—figure supplement 3*), the three S4 arginines that are known to be critical for voltage-sensing in KCNQ1 (*Shamgar et al., 2008*; *Wu et al., 2010b*). In KCNQ1 a neutral residue (Q234) is located in the canonical third arginine position (R3) of other Kv channels; therefore, we did not model the E1-R3 state. In Molecular Dynamic (MD) simulations, we found that the pore was more dilated when the VSDs were in the E1-R2 or E1-R4 states than in the E1-R1 state (*Figure 1—figure supplement 3*). These simulations suggest that KCNQ1 channels can open when the VSDs assume intermediate or activated states (*Figure 1I*, *Figure 1—figure supplement 3*). To capture these states experimentally, we engineered pairwise charge reversal mutations to arrest the VSD near the E1-R2 or E1-R4 states (*Figure 1J*, *Figure 1—figure supplement 4*). As shown previously (*Wu et al., 2010a*), for KCNQ1 channels, mutating E1 to arginine (E1R) caused a severe loss of current that was partially rescued by a charge reversing (R to E) mutation at the R2 (E1R/R2E) or R4 (E1R/R4E) positions (*Figure 1—figure supplement 4*). E1R/R2E and E1R/R4E channels were constitutively open (*Figure 1—figure supplement 4C,D*), suggesting that the VSD was trapped in states where the paired residues interact. When KCNE1 was coexpressed, the E1R/R2E currents were eliminated and the E1R/R4E currents were increased by nearly 10-fold (*Figure 1J*, *Figure 1—figure supplement 4*), suggesting that KCNE1 suppresses the intermediate-open state and potentiates the activated-open state. Importantly, we found that the abundance of E1R/R2E subunits in the cell membrane was similar when expressed alone or coexpressed with KCNE1, indicating that the inhibition of E1R/R2E currents was due to a functional effect on the intermediate-open state, not an effect on surface expression (*Figure 1—figure supplement 5B*). Altogether, the results in *Figure 1* are consistent with a model in which the VSD undergoes two sequential transitions, resting-to-intermediate and intermediate-to-activated. The first transition is sufficient for KCNQ1 to open, resulting in both intermediate-open and activated-open states. With KCNE1, the second transition is required for opening because the intermediate-open state is suppressed.

In addition to affecting KCNQ1 channel gating, KCNE1 also alters permeation and pharmacology (*Sun et al., 2012*). For example, as previously demonstrated by *Pusch et al. (2000)*, KCNQ1 channels had a higher $Rb^+/K^+$ permeability ratio than KCNQ1+KCNE1 channels (*Figure 2A,B*) and, as reported by Cohen and colleagues (*Wang et al., 2000*), KCNQ1 channels were more sensitive than KCNQ1+KCNE1 channels to the inhibitor XE991 when short duration pulses (comparable to the length of the cardiac action potential) were applied (*Figure 2—figure supplement 1*). We used the E1R/R2E and E1R/R4E mutations to examine if VSD conformation affects permeation and pharmacology. Strikingly, E1R/R4E channels had a significantly lower $Rb^+/K^+$ permeability ratio and a significantly lower apparent affinity for XE991 compared to E1R/R2E and WT KCNQ1 channels, which were similar (*Figure 2*).

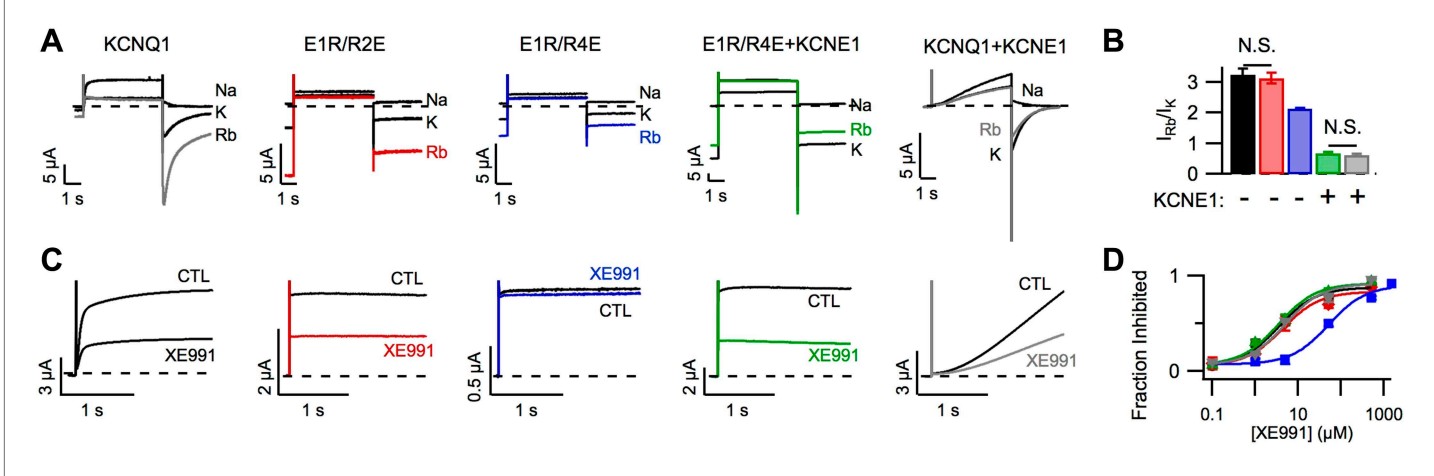

**Figure 2**. Permeation and pharmacological properties depend on VSD conformation. Currents from cells expressing WT KCNQ1 alone (KCNQ1, black), E160R/R231E (E1R/R2E, red), E160R/R237E alone (E1R/R4E, blue), E160R/R237E+KCNE1 (E1R/R4E+KCNE1, green), or WT KCNQ1+KCNE1 (KCNQ1+KCNE1, grey). (**A**) Currents from a single cell in external solutions containing 100 mM of $Na^+$, $K^+$, or $Rb^+$. The currents were elicited by first stepping the voltage to +60 mV for 5 s then to −60 mV for 3 s tails. (**B**) Averaged $Rb^+/K^+$ permeability ratios calculated by comparing the tail current amplitudes. (**C**). Currents before (CTL) and after (XE991) bath application of 5 μM XE991 in the external solution. (**D**) Fraction of original current inhibited after 2 s of depolarization vs concentration of XE991 applied shown with fits to the hill equation with a hill coefficient of 1. N.S. = not significant.

The following figure supplement is available for figure 2:

**Figure supplement 1**. Inhibition of KCNQ1+KCNE1 channels by XE991 develops slowly over time.

These results reveal that different VSD conformations yield functionally distinct open states. Furthermore, the similar properties of E1R/R2E and WT KCNQ1 suggest that the intermediate-open state is either the most populated or the most conductive open-state for WT KCNQ1 channels, a notion that is further supported by our observation that E1R/R2E currents are 2–3 times larger than E1R/R4E currents (**Figure 1J**, **Figure 1—figure supplement 4C**) despite having similar levels of membrane expression (**Figure 1—figure supplement 5A**). Relative to E1R/R4E alone, coexpression of KCNE1 with E1R/R4E significantly decreased the $Rb^+/K^+$ permeability ratio (**Figure 2A,B**) and increased the apparent affinity for XE991 (**Figure 2C,D**) indicating that, in addition to eliminating the intermediate-open state, KCNE1 alters the properties of the activated-open state. Of note, we found that E1R/R4E+KCNE1 and WT KCNQ1+KCNE1 channels displayed similar $Rb^+/K^+$ permeability ratios (**Figure 2B**) and XE991 sensitivities (**Figure 2D**), further supporting our conclusion from **Figure 1** that WT KCNQ1+KCNE1 currents are conducted by channels in the activated-open state.

In our previous studies, we have shown that KCNE1 increases the apparent affinity of KCNQ1 for the membrane lipid phosphatidylinositol 4,5-bisphosphate ($PIP_2$) (**Li et al., 2011**) and that, in KCNQ1, $PIP_2$ binding at the VSD-pore interface mediates the VSD-pore interactions that energetically couple VSD activation to pore opening (**Zaydman et al., 2013**). Taken together these findings suggest that KCNE1 affects the VSD-pore interactions. To test this hypothesis, we studied the impact of changing VSD-pore interactions directly via a point mutation of KCNQ1, F351A. F351 on S6 is highly conserved among Kv channels and participates in interactions with the S4/S5 linker that are known to be critical for VSD-pore interactions and coupling (**Lu et al., 2001**; **Tristani-Firouzi et al., 2002**; **Long et al., 2005**; **Haddad and Blunck, 2011**). Remarkably, in VCF experiments, we observed that the F351A GV was correlated with $F_{high}$ instead of $F_{main}$ (**Figure 3A**, left), revealing that, similar to KCNE1 (**Figure 1E**), F351A suppressed the intermediate-open state. As a result, F351A current onset was delayed and slowed (**Figure 3A**, right), resembling WT KCNQ1+KCNE1 current onset, as reported previously (**Boulet et al., 2007**). We also observed that, compared to WT KCNQ1, F351A caused significant changes in the $Rb^+/K^+$ permeability ratio (**Figure 3B**) and the timecourse of inhibition by XE991 (**Figure 3C**, middle, right). Of note, these properties of F351A were not identical to those of WT KCNQ1+KCNE1. However, such differences are not surprising as prior studies (**Kang et al., 2008**; **Xu et al., 2008**; **Chung et al., 2009**; **Lvov et al., 2010**; **Strutz-Seebohm et al., 2011**; **Wang et al., 2011**;

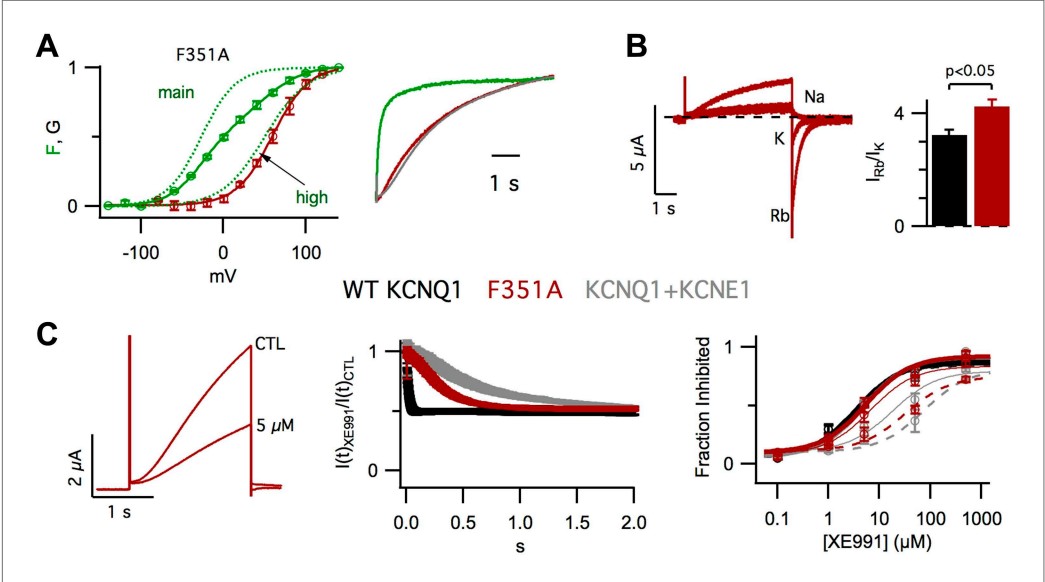

**Figure 3**. Altering VSD-pore coupling directly, by the mutation F351A, changes the gating permeation, and pharmacology of KCNQ1 channels. (**A**) VCF recordings from oocytes expressing pseudo-WT/F351A (C214A/G219C/C331A/F351A), labeled with Alexa 488 C5-maleimide. Left–GV (red) and FV (solid green) relationships with the main and high voltage FV components plotted (dotted green lines). Right–normalized fluorescence (green) and current (red) responses to a 60 mV pulse, the current from a cell expressing pseudo-WT KCNQ1+KCNE1 is shown for comparison (grey). (**B**) Left–currents from a single oocyte expressing F351A in external solutions containing 100 mM of $Na^+$, $K^+$, or $Rb^+$. Right–averaged $Rb^+/K^+$ permeability ratios for WT KCNQ1 (black) and F351A (red). (**C**) Left–currents from an oocyte expressing F351A in control and 5 μM XE991 external solutions. Middle–time dependence of XE991 inhibition—the averaged ratio of the current in 5 μM XE991 to that in control solutions is plotted vs depolarization time. Right–averaged fraction inhibited after 200 (dashed line), 500 (thin line) or 2000 (thick line) ms of depolarization is plotted vs concentration of XE991 for oocytes expressing WT KCNQ1 (black), F351A (red), or WT KCNQ1+KCNE1 (grey).

*Chan et al., 2012*; *Xu et al., 2013*) have located KCNE1 at the VSD-pore interface and have suggested that KCNE1 engages in a very broad and complex set of interactions with KCNQ1 (*Sun et al., 2012*); therefore, it would be unreasonable to expect that a single point mutation, such as F351A, would alter the VSD-pore interactions in exactly the same way as KCNE1. Nonetheless, our studies of F351A provide evidence that the VSD-pore interactions determine the permeation and pharmacological properties of the pore as well as which VSD transitions are required for the pore to open. Therefore, altering VSD-pore interactions is a single mechanism that can explain all of these effects of KCNE1.

The observation that KCNE1 suppressed the intermediate-state opening (*Figure 1*, *Figure 1—figure supplement 4*) strongly suggested that KCNE1 affects the interactions between the intermediate-state of the VSD and the pore. Does KCNE1 also affect the interaction between the activated-state of the VSD and the pore to modulate the activated-open state? In order to detect such effects of KCNE1, we used the apparent affinity of E1R/R4E for $PIP_2$ as a proxy for the strength of VSD-pore interactions in the activated-open state of KCNQ1. The rationale behind this approach was that our previous study has demonstrated that the apparent affinity for $PIP_2$ correlates with the strength of VSD-pore interaction (*Zaydman et al., 2013*). In our experiments, we used CiVSP, a voltage-sensitive lipid phosphatase (*Iwasaki et al., 2008*), to cause a rapid decrease in membrane $PIP_2$ upon depolarization (*Falkenburger et al., 2010*) and observed the resulting time-dependent decay in ionic currents resulting from net unbinding of $PIP_2$. We found that the decay of E1R/R4E+CiVSP currents were significantly slower and less severe when KCNE1 was present (*Figure 4A*), suggesting that KCNE1 increases the affinity of the activated-open state of KCNQ1 for $PIP_2$. Consistent with this idea, WT KCNQ1+KCNE1 currents, which come from activated-open states exclusively (*Figure 1*), were observed to be relatively insensitive to the activity of CiVSP as long as they were maintained in the activated-open state by sustained membrane depolarization (*Figure 4B,C*). When the channels were permitted to close, by hyperpolarizing the

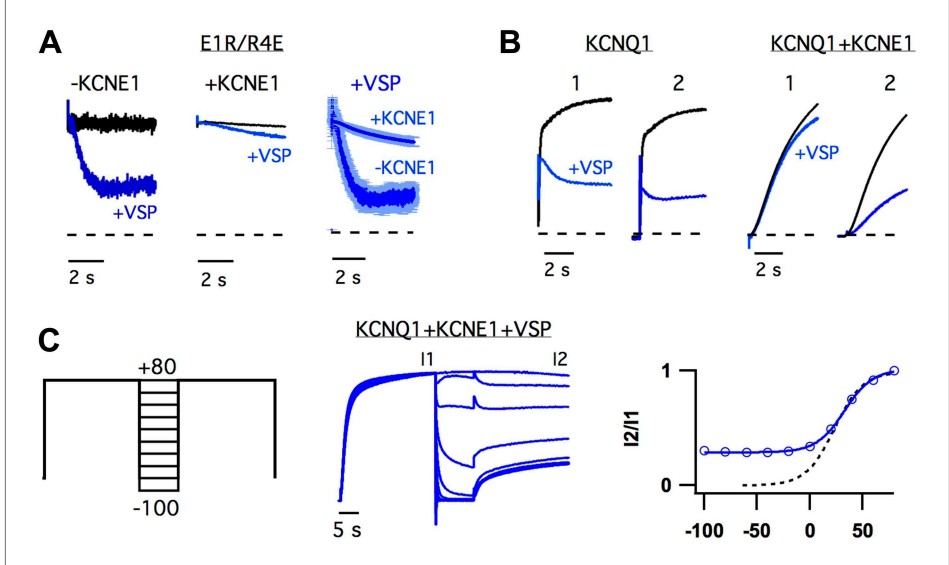

**Figure 4**. KCNE1 increases the apparent affinity of the activated-open state for $PIP_2$. (**A**) Responses of currents from oocytes expressing E1R/R4E alone (left) or E1R/R4E+KCNE1 (middle) to rapid depletion of $PIP_2$ by CiVSP (VSP, blue). The membrane voltage was pulsed to +60 mV to activate CiVSP. Currents were normalized to the value 200 ms after depolarization for comparison with currents from oocytes not expressing CiVSP (black = channel subunits alone, blue = channel subunits + CiVSP). Right—averaged current responses for oocytes expressing E1R/R4E+CiVSP (−KCNE1) or E1R/R4E+KCNE1+CiVSP (+KCNE1). (**B**) CiVSP responses (VSP, blue) of currents from oocytes expressing WT KCNQ1 or WT KCNQ1+KCNE1. Two +80 mV depolarizing pulses were applied, spaced 30 s apart (note: time scale is broken). The currents were normalized to the value 200 ms into the first depolarizing pulse for comparison with currents from an oocyte expressing channel subunits alone (black). (**C**) Left—double pulse protocol in which two 25 s depolarizing pulses were applied. In between the two pulses the membrane potential was set to various voltages for 10 s. After each sweep, the membrane potential was held at −80 mV for 300 s to deactivate CiVSP and allow for $PIP_2$ regeneration by the endogenous lipid kinases. Middle—currents from an oocyte expressing KCNQ1+KCNE1+CiVSP subjected to the voltage protocol shown and normalized to the value at the end of the first pulse (I1). Right—fraction of the first pulse current available on the second pulse (I2/I1) is plotted vs voltage of intervening 10 s (blue). The voltage-dependence of I2/I1 (solid blue line) is similar to that of the GV relationship for WT KCNQ1+KCNE1 (dotted black line) suggesting that the open probability during the 10 second interpulse determines the unbinding of $PIP_2$.

membrane potential, $PIP_2$ unbinding was facilitated as evidenced by decreased current amplitude observed with application of a subsequent depolarizing pulse (**Figure 4B,C**). Altogether these results reveal that KCNE1 causes the activated-open state to have a high apparent affinity for $PIP_2$, suggesting that KCNE1 may increase the strength of VSD-pore interactions in the activated-open state.

## Discussion

The basic experimental phenomena revealed in our study are summarized in **Figure 5A**. We observed that, in KCNQ1, VSD activation occurred in two steps, with and without KCNE1 (**Figure 1A–H**). The first step, leading to an intermediate-state, promoted the opening of homomeric KCNQ1 channels causing the GV to overlap with the first component of the FV (**Figure 1A**). KCNE1 prevented the channel from opening while the VSD is in the intermediate-state and potentiated opening while the VSD is in the activated state resulting in a GV relationship with a similar voltage-dependence as the second component of the FV (**Figure 1E**). We observed that different states of the VSD yielded functionally different open states (**Figure 2**) of the pore indicating that the unique sets of interactions with different conformations of the VSD determine the probability of pore opening and selects for different open-pore conformations. In agreement with this interpretation, changing the VSD-pore interactions directly, via the mutation F351A, suppressed the intermediate-open state and changed the properties of the open channel (**Figure 3**). The ability of this single point mutation at the S4–S5/S6

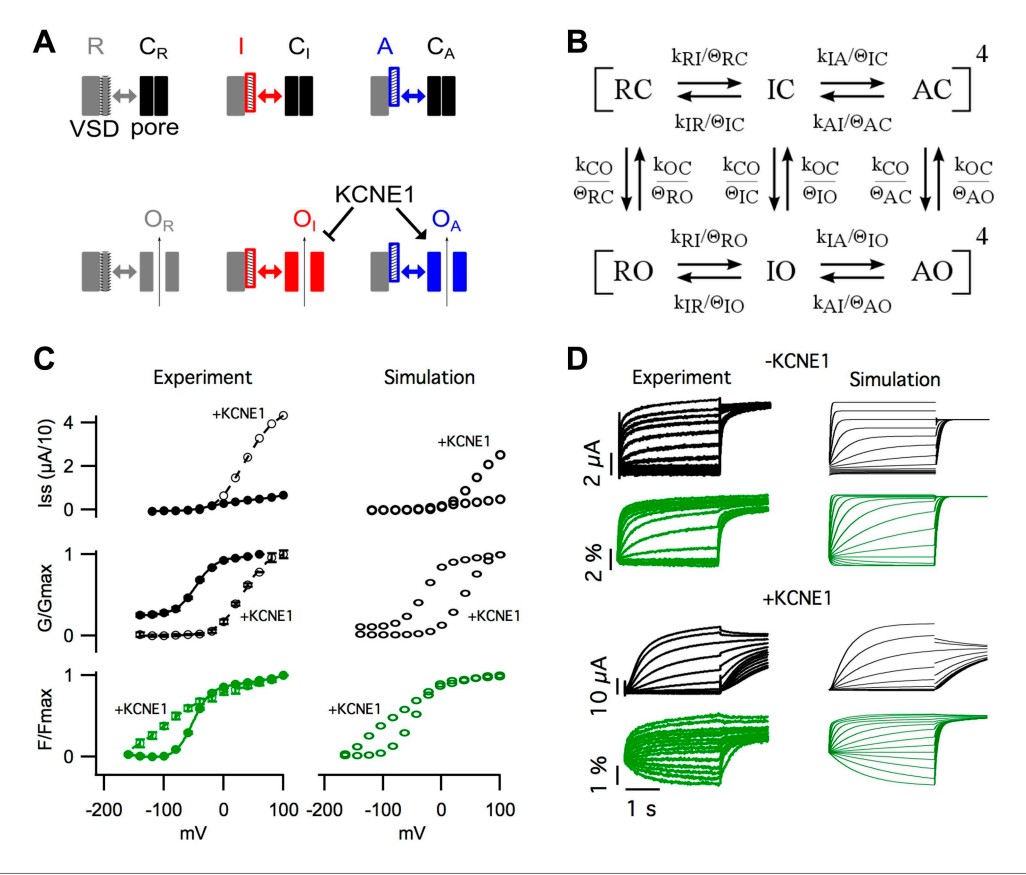

**Figure 5**. Modeling the effects of KCNE1. (**A**) Cartoon illustrating the observed effects of KCNE1 on KCNQ1. The VSDs transit between resting (R, grey patterned), intermediate (I, red patterned), and activated (A, blue patterned) states. Each VSD conformation has unique interactions (double arrow) with the closed (C) and open (O) conformation of the pore. KCNE1 suppresses the intermediate-open ($O_I$) state and modulates the activated-open ($O_A$) states. (**B**) Kinetic model of KCNQ1 channel gating where the k parameters are the intrinsic transition rates of the VSD and the pore, and the θ parameters explicitly represent VSD-pore interactions. Fourth power notation ([ ]⁴) indicates that the model includes four VSDs. (**C** and **D**) Experimental data and model simulations with (+KCNE1) and without KCNE1. Steady-state current- (**C**, top), conductance- (**C**, middle), and fluorescence- (**C**, bottom) voltage relationships. (**D**) Current (black) and fluorescence (green) responses at various voltages.

The following figure supplements are available for figure 5:

**Figure supplement 1**. Scheme for voltage-depededent gating.

**Figure supplement 2**. Balanced gating model showing transition rates.

**Figure supplement 3**. Desriptions, values and units for free parameters of gating model.

**Figure supplement 4**. Descriptions, values, and units for constant parameters used in gating model simulation.

interface to change the same variety of properties as KCNE1 suggests that a single mechanism for the effects of KCNE1 could be to alter the VSD-pore interactions.

Experimentally, two effects of KCNE1 on VSD-pore interactions were observed. (1) KCNE1 prevented the channel from opening when the VSD was in the intermediate-state suggesting that KCNE1 changes the interactions between the intermediate-state of the VSD and the pore (*Figure 1*). (2) KCNE1 greatly increased the apparent affinity of the activated-open state for $PIP_2$ suggesting that KCNE1 stabilized the activated-open state interactions (*Figure 4*). Using kinetic modeling, we sought to test if these experimentally observed effects of KCNE1 on VSD-pore interactions are sufficient to

explain the many effects of KCNE1 on KCNQ1 activation gating. Previous Kv channel gating models assume that all four VSDs (*Zagotta et al., 1994*) or each VSD (*Horrigan et al., 1999*) must fully activate before the pore can open. Therefore, we could not use these established models as they are not able to represent the coupling between three different states of the VSD and the pore. We developed a new gating model that better reflects the understanding that the VSD and pore can fold and function independently (discussed in our previous review [*Zaydman and Cui, 2014*]), but couple their motions through state-dependent interactions.

In our gating model, the VSDs can occupy resting (R), intermediate (I), or activated (A) states, and the pore can occupy closed (C) or open (O) states (*Figure 5B*, *Figure 5—figure supplement 1*). Assuming that the intrinsic activation of each VSD is identical and non-cooperative, as assumed in the ZHA model (*Zagotta et al., 1994*), the different combinations of the states of the four VSDs and the pore give rise to only 30 possible channel states. Two sets of parameters, k's and θ's, determine the transitions among these channel states (*Figure 5B*, *Figure 5—figure supplement 2,3*). The 'k' parameters represent the intrinsic tendencies of the pore to open and close and a VSD to undergo its transitions, and would be measured directly if the VSD and pore could be decoupled entirely. The k parameters concerning VSD transitions (kRI, kIR, kIA, kAI) are assumed to be exponentially dependent on voltage according to *Equation 1*. On the other hand, the k parameters concerning pore transitions (kCO and kOC) are assumed to be voltage-independent (i.e., constant).

$$k_{ij} = k_{ij}0 * \exp\left(z_{ij} * F * V / (R * T)\right) \tag{1}$$

where: $k_{ij}$ is the voltage-dependent rate of transition from VSD state i to VSD state j, $k_{ij}0$ is the rate of the ij transition at 0 mV, $z_{ij}$ is the valence of the ij transition, F is the faraday constant, V is voltage, R is the universal gas constant, T is the absolute temperature.

The 'θ' terms explicitly represent the net effect of all VSD-pore interactions within each channel state. For example, $\theta_{IC}$, represents the net stabilization of the intermediate-closed state due to all interactions between the intermediate-state of the VSD and the closed-state of the pore. If $\theta_{IC}$ is greater than 1, these interactions will slow transitions away from the intermediate-closed state. The full model is shown in *Figure 5—figure supplement 2* and a shorthand version appears in *Figure 5B*. The main difference between our model and previous allosteric gating models is that the reference state is an imaginary state in which the two domains are completely isolated (i.e., decoupled) from each other, rather than using transitions among the resting states as a reference. The advantage of our approach is that each parameter has an intuitive physical meaning: the k parameters represent the net effect of all interactions within a state of the VSD or the pore, and the θ parameters represent the net effect of all interactions between the VSD and the pore in a specific channel state.

In order to parameterize our model, we started with parameter values from two previous KCNQ1 channel gating models (*Silva and Rudy, 2005*; *Zaydman et al., 2013*) and were able to reasonably replicate the experimentally observed KCNQ1 channel gating behavior (*Figure 5C,D*) with some adjustments to the parameter values to account for the differences in the schemes of these models (*Figure 5—figure supplement 3*). In order to simulate the effects of KCNE1 on VSD-pore interactions, we made only two changes to the VSD-pore interactions, as suggested by experimental results (*Figure 5—figure supplement 3*). (1) Experimentally, when KCNE1 was coexpressed, the pore remained closed while the VSD was in the intermediate state (*Figure 1E*, *Figure 1—figure supplement 4*). Thus, to model KCNE1, we strengthened the intermediate-closed state interaction, $\theta_{IC}$. (2) Also, in experiments, KCNE1 increased the activated-open state affinity for PIP$_2$ (*Figure 4*). Accordingly, we strengthened the activated-open state interaction, $\theta_{AO}$. Making only these two changes mimicked all of the effects of KCNE1 on KCNQ1 activation gating (*Figure 5C,D*). The GV shifted to more depolarized voltages correlating with $F_{high}$ because the intermediate-closed state interactions became more energetically favorable than the intermediate-open state interactions, preventing pore opening until a more depolarized voltage-range where the intermediate-to-activated transition occurred. In the absence of intermediate-state opening, the resting-to-intermediate transition occurred among closed states and introduced the characteristic delay of current onset of several hundred milliseconds. The maximal current amplitude was increased several fold due to stabilization of the activated-open state. Furthermore, the stabilization of the intermediate-closed state reduced the current near physiological resting voltages and caused a left shift in main component of the FV curve, two phenomena observed experimentally (*Figure 5C*). The kinetics and steady-state gating behavior predicted by our model were not quantitatively identical to

those in experiments; such discrepancies were expected due to several overly simplistic assumptions that we used to limit the number of states in our model. Particularly, we assumed that $PIP_2$ binding was saturated, that all open states had identical conductance, and that inactivated states did not exist. However, previous studies highlight that KCNQ1 is not saturated with $PIP_2$ and that KCNE1 dramatically increases $PIP_2$ binding (*Li et al., 2011*), that KCNE1 increases the apparent single channel conductance (*Sesti and Goldstein, 1998*; *Yang and Sigworth, 1998*), and that KCNE1 prevents the observation of a partially inactivated state (*Pusch et al., 1998*; *Tristani-Firouzi and Sanguinetti, 1998*), all of which may contribute to gating and macroscopic current amplitude. Revision of our model to include the influence of $PIP_2$ binding and the different properties of intermediate- and activated-open states will require additional studies to better define these properties. Despite these limitations, it is remarkable that we were able to capture all the major effects of KCNE1 on the activation gating of KCNQ1. This model illustrates that the effects of KCNE1 on VSD-pore interactions are sufficient to explain how KCNE1 affects the activation gating of KCNQ1 without any additional effects on VSD activation or pore opening. Furthermore, as our experimental data demonstrate that VSD-pore interactions determine the permeation and pharmacological properties (*Figures 2,3*), our proposed mechanism, that KCNE1 affects VSD-pore interactions, provides a relatively complete explanation for how KCNE1 affects KCNQ1.

Central to understanding the modulation of KCNQ1 by KCNE1 is a longstanding controversy regarding which stoichiometries of KCNQ1:KCNE1 may exist in the fully assembled channel. Several groups have argued that association of KCNE1 with KCNQ1 dimers during an early stage of biogenesis leads to a fixed 4:2 KCNQ1:KCNE1 stoichiometry and breaks the fourfold symmetry of the channel (*Wang and Goldstein, 1995*; *Chen et al., 2003*; *Morin and Kobertz, 2008*; *Plant et al.,2014*). Other groups have argued that various stoichiometries, 4:0–4 KCNQ1:KCNE1, are possible, depending on the relative levels of subunit expression (*Blumenthal and Kaczmarek, 1994*; *Cui et al., 1994*; *Wang et al., 1998*; *Nakajo et al., 2010*; *Li et al., 2011*). In the present study, we coinjected oocytes with KCNQ1 and KCNE1 transcripts at a 1:1 ratio because previous work, from our lab (*Li et al., 2011*) and others (*Nakajo et al., 2010*), demonstrated that this ratio is sufficient to saturate the functional effects of KCNE1 on KCNQ1. In our modeling, we assume that coexpression of KCNE1 affects all four subunits identically as if the channel were saturated by KCNE1, that is, with a 4:4 stoichiometry. This assumption may represent yet another reason why the simulated and experimental gating behavior are not quantitatively identical. Furthermore, it is important to ask, given the ongoing controversy regarding stoichiometry, if the presence of multiple populations of channels with different stoichiometries could complicate our interpretation of our data. Fortunately, the F351A mutation demonstrates that the major finding of our study—the ability to couple pore opening to different transitions within the VSD—is intrinsic to the KCNQ1 subunit and observed even in the absence of the KCNE1 subunit.

In the voltage-gated ion channel field, VSD-pore coupling, aka electromechanical coupling, has been a loosely defined term referring to the experimental observation that pore opening is more likely when the VSDs are activated at depolarized voltages. In our model of voltage-dependent gating, coupling is represented in a very different way than in the previously established models. The landmark linear models of Shaker Kv channels (*Schoppa et al., 1992*; *Zagotta et al., 1994*), which require that all four VSDs be activated before the pore can open, do an excellent job replicating the gating of Shaker Kv channels, but do not explicitly define coupling. Another landmark model, the Horrigan-Cui-Aldrich (HCA) model (*Horrigan et al., 1999*), permits pore-opening without prior activation of all VSDs. In the HCA model, coupling is quantitatively defined by a single term, D, representing how much the closed-open equilibrium of the pore is biased towards open when a single VSD is transitioned from its fully resting to its fully activated state. Our model further develops from these two previous models. Like the shaker model, we assumed two transitions occurring sequentially and independently within each VSD. Similar to the HCA model, we assumed that pore opening can occur with the four VSDs in any combination of states. In our model, we decomposed D into its elementary components, that is, VSD-pore interactions at each channel state. As a result, each parameter (k, θ) in our model represents a specific set of interactions that exist at the same time in the physical world. The relative differences in the strengths of all these state-dependent VSD-pore interactions lead to the experimental observation that a change in VSD conformation leads to a change in the probability of pore opening, that is, coupling. It is important to understand that coupling is a result of all of these state-dependent interactions and a perturbation of any of these interactions may alter the coupling. These points are illustrated by our modeling of the KCNE1 effect on KCNQ1 channel gating where strengthening the intermediate-closed interactions caused an apparent decoupling of pore-opening

from the resting-to-intermediate transition of the VSD, that is, the open probability was no longer increased by the transition of the VSD to the intermediate-state at intermediate voltages. Alternatively, weakening the intermediate-open state interactions would also decouple opening from the resting-to-intermediate state of the VSD; however, this would not reproduce the leftward shift of the first FV component ($F_{main}$) that we observed when KCNE1 was expressed (*Figure 5C*).

The topic of how KCNE1 modulates KCNQ1 has been studied for many years. Early studies by Goldstein and colleagues argued that KCNE1 intercalates deeply into the pore and directly lines the permeation pathway (*Goldstein and Miller, 1991*; *Wang et al., 1996*; *Tai and Goldstein, 1998*). Later, the Kass and George labs argued that KCNE1 associates outside the pore and modulates the pore properties through an allosteric mechanism (*Kurokawa et al., 2001*; *Tapper and George, 2001*). Then, the efforts of several labs have detected probable interactions between the extracellular (*Xu et al., 2008*; *Chan et al., 2012*), transmembrane (*Tapper and George, 2001*; *Chung et al., 2009*; *Lvov et al., 2010*), and intracellular (*Haitin et al., 2009*) regions of KCNQ1 and KCNE1. These data have been used to build homology models of the KCNQ1+KCNE1 channel (*Kang et al., 2008*; *Lundby et al., 2010*; *Xu et al., 2013*), all of which place KCNE1 in a cleft between a VSD and the pore where it participates in a broad set of interactions with KCNQ1. Even after all of these excellent studies, there was a lack of a biophysical mechanism to explain how these interactions alter the gating, permeation, and pharmacology of KCNQ1. Prior biophysical studies, which have focused on how KCNE1 slows current onset, have come to conflicting conclusions that KCNE1 either slows VSD activation (*Nakajo and Kubo, 2007*; *Ruscic et al., 2013*) or pore opening (*Rocheleau and Kobertz, 2008*; *Osteen et al., 2010*). We believe that the interpretations of these data were limited by several missing pieces of information revealed in the present study: (1) VSD activation occurs through a stable intermediate state, (2) full activation of the VSD is not necessarily required to open the pore, and (3) the distinct interactions between different states of the VSD and the pore determine both the probability of opening and the open-state conformation. Also, it is likely that these previous studies did not consider an effect on the state-dependent VSD-pore interactions because such interactions are not explicitly represented in previously established gating models.

Recently, the Larsson and Kass labs reported the presence of a second fluorescence transition in KCNQ1+KCNE1 channels, which they attributed to pore opening (*Osteen et al., 2010*, *2012*) or a concerted step in which pore opening and further S4 movement occur simultaneously (*Barro-Soria et al., 2014*). From these observations, they concluded that KCNE1 changes the number of subunits that must be activated before pore opening can occur leading to a right shifted GV and a delay in current onset. This model is most similar to ours in that KCNE1 is changing the relationship between VSD activation and pore opening; however, our model is different in several regards. We believe that the second FV component reports on an intrinsic transition of the VSD rather than the pore opening step. This assumption was based on our observation that homomeric KCNQ1 channels also exhibited a second FV component, with similar properties to that of KCNQ1+KCNE1, which occurred in a range of voltages that was more positive than required for pore opening (*Figure 1A–H*, *Figure 1—figure supplement 1*). Thus, in our model, channel opening requires an additional transition within a single VSD, rather than the activation of additional subunits. A natural consequence of such gating behavior is the existence of both intermediate-open and activated-open states, which could be detected experimentally and are of great functional importance as the different open states have different conductive and pharmacological properties (*Figure 2*).

## Materials and methods

### Mutagenesis

Point mutations were engineered using overlap extension and high-fidelity PCR. Each mutation was verified by DNA sequencing. cRNA was synthesized using the mMessage T7 polymerase kit (Applied Biosystems).

### Oocytes expression

Pieces of ovarian lobes were excised from *Xenopus laevis* by laparotomy. Stage V or VI oocytes from *X. laevis* were isolated by collagenase (Sigma Aldrich, St Louis, MO) digestion. 9.2 ng of KCNQ1 cRNA was microinjected with or without 2.3 ng of KCNE1 cRNA (using the Drummond Nanoject, Broomall, PA) into each oocyte. For CiVSP expression, 2.3 ng of CiVSP cRNA was coinjected. Injected cells were incubated at 18°C for up to 7 days before recording in ND96 solution (96 mM NaCl, 2 mM KCl, 1.8 mM CaCl2, 1 mM MgCl2, 5 mM HEPES, pH 7.6).

## Electrophysiology

### Two-electrode voltage clamp

Microelectrodes were pulled with resistances between 0.3 and 3 MΩ and filled with 3 M KCl solution. Recordings were performed in ND96 solutions unless otherwise indicated. Whole-oocyte currents in response to applied voltage steps were amplified using the CA-1B (Dagan, Minneapolis, MN) amplifier in two-electrode voltage clamp mode and digitized using the HEKA EPC10 (HEKA, Germany) AD/DA board, sampled at 1 KHz and recorded using the Patchmaster (HEKA) software.

### Voltage clamp fluorometry

Oocytes were labeled with 10 μM Alexa 488 C5-maleimide or Alexa 546 C5-maleimide (Molecular Probes, Eugene, OR) in high $K^+$ solution (98 mM KCl, 1.8 mM CaCl2, 5 mM HEPES, pH 7.6) for 45 min on ice. After labeling, the cells were washed with ND96 and kept on ice until recording. Recordings were performed in ND96 solution. Fluorescence emission from the sample was focused onto a Pin20A photo-diode (OSI Optoelectronics), amplified by an EPC10 (HEKA) patch amplifier, analog filtered at 200 Hz, sampled at 1 KHz, and recorded simultaneously with whole oocyte currents. A FITC filter cube (Leica, Germany) was used for Alexa 488 labeled cells and a rhodamine cube (Leica) was used for cells labeled with Alexa 546.

### Voltage-protocols

Holding potential was set to −80 mV throughout. Voltage steps were applied to elicit current and fluorescence signals. Tail potentials were +60 mV for VCF experiments and measuring IV curves, −60 mV for $Rb^+/K^+$ permeability experiments, and −40 mV for XE991 experiments.

## Molecular dynamics (MD) simulations

In our previous work (Kasimova et al.,), we have built models of the Kv7.1 activated/open and resting/closed states using homology modeling (*Eswar et al., 2007*) with the Kv1.2 crystal structure in its activated/open state (pdb code 3LUT [*Chen et al., 2010*]), α, and in its δ conformational state (*Delemotte et al., 2011*) as templates. Here, we applied a similar protocol to prepare a model of the Kv7.1 intermediate state. Each state is characterized by a unique set of interactions within the VSD. In particular, E160 (E1) forms a salt bridge with R237 (R4) in the activated/open or with R228 (R1) in the resting/closed states (*Figure 1I*, S3a). We assumed that, when the VSD is intermediate, E1 interacts with the residue located in between of R4 and R1, namely R231 (R2). Based on this assumption, the γ conformation of Kv1.2 (*Delemotte et al., 2011*) with the E1-R2, was considered as a template for the Kv7.1 intermediate state model. This model was further embedded in a palmitoyl-oleyl-phosphati-dylcholine (POPC) hydrated bilayer and immersed in a 150 mM $K^+Cl^-$ solution. Due to the importance of $PIP_2$ for Kv7.1 function (*Loussouarn et al., 2003*; *Eswar et al., 2007*), four molecules of this lipid were placed at the channel's intrasubunit sites located at the interface between the voltage sensor and the pore (*Zaydman et al., 2013*; Kasimova et al.,).

The MD simulations were performed using NAMD (*Smart et al., 1996*). Langevin dynamics was applied to keep the temperature (300 K) and the pressure (1 atm) constant. The time-step of the simulations was 2.0 fs. The equations of motion were integrated using a multiple time-step algorithm. Short- and long-range forces were calculated every 1 and 2 time-steps respectively. Long-range electrostatics was calculated using Particle Mesh Ewald (PME). The cutoff distance of short-range electrostatics was taken to be 11 Å. A switching function was used between 8 and 11 Å to smoothly bring the vdW forces and energies to 0 at 11 Å. During the calculations, chemical bonds between hydrogen and heavy atoms were constrained to their equilibrium values. Periodic boundary conditions were applied.

The protein backbone was constrained during 100 ns allowing $PIP_2$ to sample possible interactions with the channel's positive residues. Based on this trajectory, the time evolution of salt bridges forma-tion was monitored. Several residues of Kv7.1 interacted with the lipid headgroups temporarily, revealing different configurations of the system where corresponding salt bridges were either formed or broken. In total, for the activated/open, intermediate and resting/closed states we have identified 9, 8 and 8 the most frequent configurations respectively. These were considered as starting points for the final equilibration step, involving gradual release of the protein backbone and subsequent relaxa-tion of the entire system during 100 ns for each. For all the trajectories, the root mean square deviation (RMSD) from the initial structure reached a plateau starting from ~50 ns. The simulation stretch from 50 to 100 ns was used for further analysis.

In order to estimate a degree of the Kv7.1 pore dilation at the intercellular gate level, we applied HOLE (*Smart et al., 1996*). 50 conformations of Kv7.1 spread equidistantly along the last 50 ns were extracted from each MD trajectory. For these conformations, the pore radius along the axis normal to the membrane (Z) was calculated. The obtained profiles were considered to estimate an average profile and error bars (SD) for each of the channel states.

To analyze the salt bridge formation between $PIP_2$ and Kv7.1, we measured the minimal distance between the nitrogen atoms of arginine and lysine charged groups and the oxygen atoms of the $PIP_2$ phosphates. The salt bridges were assumed formed if the calculated distance was less than 3.2 Å. The probabilities of salt bridge formation were simultaneously estimated for four subunits of the channel as a ratio between the number of frames with a formed salt bridge to its total number. The error bars correspond to a standard deviation (SD) calculated between values obtained from several MD runs.

### Data analysis

Relative conductance-voltage (GV) relationships were generated by estimating the instantaneous tail current values following test pulses to various voltages and normalizing to the value following the highest voltage test pulse. For calculation of relative fluorescence changes, a baseline fluorescence was extrapolated by fitting a line to the fluorescence at the holding potential during the 2 s preceding application of the voltage pulse. ΔF/F was calculated as (F(t)-baseline(t))/baseline(t), where F(t) is the raw fluorescence intensity at time t and baseline(t) is the extrapolated baseline value at time t. Fluorescence voltage-relationships were derived by normalizing the ΔF/F value at the end of a four second test pulse to various voltages to the value of the highest voltage test-pulse. FV and GV curves were fits with one or the sum of two Boltzmann equations in the form $1/(1 + \exp(-z*F*(V - V_{1/2})/RT))$ where z is the equivalent valence of the transition, $V_{1/2}$ is the voltage at which the transition is half maximal, R is the gas constant, T is absolute temperature, F is the Faraday constant and V is the voltage. FV curves were derived from the value at the end of the test pulse, GV curved were derived from estimating the instantaneous tail current amplitude.

### Statistics

All averaged data reflects n = 6 or more from at least two batches of oocytes. Pairwise comparisons were achieved using Student's *t* test, multiple comparisons were performed using an ANOVA with Tukey's Post-Hoc Test. All error bars represent standard error mean.

### Solutions and chemicals

100 mM $Na^+$ (96 mM NaCl, 4 mM KCl, 1.8 mM $CaCl_2$, 1 mM $MgCl_2$, 5 mM HEPES) 100 mM $K^+$(100 mM KCl, 1.8 mM $CaCl_2$, 1 mM $MgCl_2$, 5 mM HEPES) 100 mM Rb (96 mM RbCl, 4 mM KCl, 1.8 mM $CaCl_2$, 1 mM $MgCl_2$, 5 mM HEPES) ND96 (96 mM NaCl, 4 mM KCl, 1.8 mM $CaCl_2$, 1 mM $MgCl_2$, 5 mM HEPES) XE991 from Sigma Aldrich. Alexa fluors from Molecular Probes.

## Acknowledgements

We thank Dr Y Okamura (Osaka University) for the CiVSP clone and Dr S Goldstein for the KCNQ1 clone. We also thank Drs A Krumholz, HP Larsson, R Barro, and A Federman for review of the manuscript. This work was supported by NIH Grants R01-HL70393 and R01-NS060706, and National Science Foundation of China Grant 31271143 and Major International Joint Research Program Fund of China 81120108004 (to Jianmin Cui), and American Heart Association predoctoral fellowship 11PRE5720009 and NIH Training Grant T32 HL007873 (to MAZ). We acknowledge PRACE for awarding us access to resource CURIE based in France at the TGCC and SUPERMUC based in Germany at the LRZ.

## Additional information

### Funding

| Funder | Grant reference number | Author |
|---|---|---|
| National Institutes of Health | R01-HL70393 and R01-NS060706 | Jianmin Cui |
| National Natural Science Foundation of China | 31271143 | Jianmin Cui |

| Funder | Grant reference number | Author |
|---|---|---|
| National Natural Science Foundation of China | Major International Joint Research Program Fund of China 81120108004 | Jianmin Cui |
| American Heart Association | 11PRE5720009 | Mark A Zaydman |
| National Institutes of Health | T32 HL007873 | Mark A Zaydman |

The funders had no role in study design, data collection and interpretation, or the decision to submit the work for publication.

## Author contributions

MAZ, MAK, Conception and design, Acquisition of data, Analysis and interpretation of data, Drafting or revising the article; KMF, Conception and design, Acquisition of data; ZB, MT, Acquisition of data, Analysis and interpretation of data, Drafting or revising the article; PH, Designed and measured the time dependence and steady-state XE991 inhibition, Conception and design, Acquisition of data; HEK, Performed Western Blotting to assess membrane expression of WT and mutant channels, Conception and design, Acquisition of data; HL, Harvested oocytes, Performed biotinylation of membrane proteins, Acquisition of data, Contributed unpublished essential data or reagents; GZ, Designed assay of membrane expression, harvested oocytes for protein expression, Conception and design, Acquisition of data; JS, Acquisition of data, Drafting or revising the article; JC, Conception and design, Analysis and interpretation of data, Drafting or revising the article

## Ethics

Animal experimentation: All animal handling in this study was carried out in accordance with a protocol approved by the Animal Studies Committee at Washington University in St Louis (Protocol #20130060). Great care and attention was spent to ensure minimize suffering and hardship.

# Additional files

## Major dataset

The following previously published datasets were used:

| Author(s) | Year | Dataset title | Dataset ID and/or URL | Database, license, and accessibility information |
|---|---|---|---|---|
| Chen X, Wang Q, Ni F, Ma J | 2010 | A Structural Model for the Full-length Shaker Potassium Channel Kv1.2 | http://www.pdb.org/pdb/explore/explore.do?structureId=3LUT | Publicly available at RCSB Protein Data Bank. |
| Long SB, Tao X, Campbell EB, MacKinnon R | 2007 | Shaker family voltage dependent potassium channel (kv1.2-kv2.1 paddle chimera channel) in association with beta subunit | http://www.pdb.org/pdb/explore/explore.do?structureId=2R9R | Publicly available at RCSB Protein Data Bank. |
| Bagneris C, Decaen PG, Hall BA, Naylor CE, Clapham DE, Kay CWM, Wallace BA | 2013 | Open-form NavMS Sodium Channel Pore (with C-terminal Domain) | http://www.pdb.org/pdb/explore/explore.do?structureId=3ZJZ | Publicly available at RCSB Protein Data Bank. |
| Zhou Y, Morais-Cabral JH, Kaufman A, MacKinnon R | 2001 | Potassium Channel KcsA-Fab complex in high concentration of K+ | http://www.pdb.org/pdb/explore/explore.do?structureId=1K4C | Publicly available at RCSB Protein Data Bank. |
| Payandeh J, Gamal El-Din TM, Scheuer T, Zheng N, Catterall WA | 2012 | Crystal structure of the NavAb voltage-gated sodium channel (wild-type, 3.2 A) | http://www.pdb.org/pdb/explore/explore.do?structureId=4EKW | Publicly available at RCSB Protein Data Bank. |
| Zhang X, Ren WL, DeCaen P, Yan CY, Tao X, Tang L, Wang JJ, Hasegawa K, Kumasaka T, He JH, Wang JW, Clapham DE, Yan N | 2012 | Crystal structure of NavRh, a voltage-gated sodium channel | http://www.pdb.org/pdb/explore/explore.do?structureId=4DXW | Publicly available at RCSB Protein Data Bank. |

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
