## [Decision Letter]

Thank you for sending your work entitled “KCNE1 modulates KCNQ1 channel gating,
permeation and pharmacology through a single mechanism” for consideration at
*eLife*. Your article has been favorably evaluated by John Kuriyan
(Senior editor), Richard Aldrich (Reviewing editor), and 3 reviewers.

The Reviewing editor and the reviewers discussed their comments before we reached this
decision, and the Reviewing editor has assembled the following comments to help you
prepare a revised submission.

The biophysical and pharmacological properties of KCNQ1 K+ channels are radically
altered by co-assembly with KCNE1 accessory beta-subunits. In this study, the authors
use voltage-clamp fluorometry (VCF) to show that activation of the voltage sensor domain
(VSD) of KCNQ1 channels occurs in two resolvable steps. Both the intermediate and
fully-activated states of the VSD result in KCNQ1 channel opening, and these states have
different permeation and pharmacological properties. The intermediate-open state was
absent when KCNQ1 was co-expressed with its beta-subunit KCNE1, suggesting that
accessory subunits changes how the VSD and pore interact. The gating of F351A KCNQ1
channels in many ways mimics KCNQ1/KCNE1 channel gating, supporting the main hypothesis.
Finally, a kinetic model was developed that can account for many of the effects of KCNE1
on VSD-pore interactions. These findings provide important and novel insights into the
mechanism of IKs channel gating as well as a framework for interpreting KCNQ effects
that is novel, well substantiated, conceptually simple, and thermodynamically
plausible.

The authors need to more clearly delineate and critically evaluate which aspects of the
gating and permeation/conductance differences between the two channels their model does
or does not fully describe.

The following issues must be addressed in a revision:

1) The main conclusion, that KCNE1 alters KCNQ1 gating by a single mechanism (altered
VSD-pore coupling; modeled by only two changes to a kinetic model) is attractive due to
its apparent simplicity; however in reality the effects of the accessory subunit are
more complex than just shifting the G-V and slowing activation kinetics. KCNQ1 channels
inactivate, whereas KCNQ1/KCNE1 channels do not. Some of the deviation between the data
and the predictions of the model may be due to KCNQ1 inactivation, which is not
detectable using voltage-activation protocols when KCNE1 is present – something
the authors might want to consider in the interpretation of their data.

2) The single channel properties of KCNQ1 channels differ from KCNQ1/KCNE1 channels. The
model should be modified to account for these important differences in channel
properties. One would expect for example two discernible channel open states for KCNQ1
(corresponding to the intermediate-open (O1) and activated-open (O2) states), whereas
KCNQ1/KCNE1 channels would have only one open state, with properties similar to the O2
state for KCNQ1 channels. It is stated that “The maximal current amplitude was
increased several fold due to stabilization of the activated-open state”.
Presumably, this means only a change in Po, whereas experimental data indicate that
KCNQ1/KCNE1 currents are larger mainly due to an increase in single channel conductance
(Yang and Sigworth, 1998. J Gen Physiol 112:665).

3) The data presented in Figure 3 and
corresponding text are incomplete and their interpretation (that it indicates increased
“strength” of VSD-pore coupling in presence of KCNE1) is not convincing.
F351A channels have many properties similar to KCNQ1/KCNE1 channels (especially G-V
matching Fmain-V relationship), but the relative Rb/K ion permeation and pharmacology
are not the same. The authors state that F351A like KCNE1 cause significant changes in
the Rb/K permeability ratio (Figure 3), however
this change is in the opposite direction from KCNE1 (Figure 2). This KCNQ1 mutant with KCNE1-like activation kinetics should have
a significantly lower (not a higher) permeability ratio if it is mimicking KCNQ1/KCNE1
channels. This should be made clear. Similarly, F351A affects XE991 pharmacology, but
this is unlike the effect of KCNE1. These experiments need to be expanded (e.g.,
dose-response to XE991; differences in compound sensitivity requires more than a single
concentration) to more quantitatively describe the effects of KCNE1. In addition, the
rubidium data and pore dilation hypothesis might be better resolved using external
cesium, which is larger, yet permeates homotetrameric KCNQ1 channels.

4) Several figures use whole cell current amplitude as a proxy for channel gating.
Authors should address the possibility of surface expression level differences with
different proteins, KCNQ mutations and plus/minus KCNE.

5) CiVSP experiments need to be better explained and analyzed. The protocol for CiVSP
activation is not explained. There is no analysis or evidence of repeated trials, and
the logic is confusing. In the text the claim is made that Figure 3 demonstrates that KCNE1 increases the open state affinity
for PIP2. This claim is not well backed up. Explaining why a lack of effect of CiVSP
demonstrates high affinity rather than no interaction with PIP2 could be helpful.

6) The modeling could be explained better. The modeled effect of KCNQ invokes a 10 fold
stabilization of IC and AO states with no alteration of transition barrier heights. In
effect, this results in decreased coupling to the IO state (because the RO is rarely
occupied). The implied change in coupling could be more thoroughly discussed, as this is
the major intellectual contribution of this study.

7) While it is mentioned that the kinetics of the model don't entirely match the
data, it appears that a major feature of KCNQ gating is missed by the model. The
“tau slow” of KCNQ (quantified and made note of in the discussion
surrounding Figure 1) is proposed to be the
transition from an IO state to an AO state. The existence of this slow component is a
pillar of their mechanism involving multiple classes of open states. This slow component
seems to be missing from the KCNQ1 modeled kinetics.

8) Statistics. Students T test was used without explanation of why such a parametric
test is appropriate.

9) Figure 1—figure supplement 4.
Analyzing normalized GVs here would be more informative than just the IVs in panels
(c),(d).

10) Figure 2: Explain voltage protocols used
more completely. If KCNQ is slowly transiting from an IO state to an AO state
wouldn't the Rb/K permeability ratio be time and voltage dependent?

11) The authors should discuss/hypothesize how two KCNE1 subunits fit into their
modeling (Figure 4) given the recent paper by the
Goldstein lab unequivocally showing that the stoichiometry of the complex is 4:2
Q1:E1.

12) XE991 Pharmacology: The Wang et al. paper showed that KCNQ1/KCNE1 channels are
10-fold less sensitive than KCNQ1 channels; however, in Figure 2 wild type KCNQ1/KCNE1 channels do not appear to be 10-fold less
sensitive to the drug. Of all the currents shown, these are the smallest, which makes us
wonder if these are endogenous xKCNQ1 channels assembling with exogenously expressed
KCNE1 subunits. Besides the inconsistency with wild type, the E1R/R4E+KCNE1 mutant
should have similar “inhibition” to E1R/R4E. It appears that the E1R/R4E
mutant is modestly activated by XE991.

---

## [Author Response]

*1) The main conclusion, that KCNE1 alters KCNQ1 gating by a single mechanism
(altered VSD-pore coupling; modeled by only two changes to a kinetic model) is
attractive due to its apparent simplicity; however in reality the effects of the
accessory subunit are more complex than just shifting the G-V and slowing activation
kinetics. KCNQ1 channels inactivate, whereas KCNQ1/KCNE1 channels do not. Some of the
deviation between the data and the predictions of the model may be due to KCNQ1
inactivation, which is not detectable using voltage-activation protocols when KCNE1
is present – something the authors might want to consider in the
interpretation of their data*.

The reviewers are right that besides the reasons that we have described, the presence of
inactivation gating likely also contributes to the discrepancy between the model
kinetics and the experimental data. We have added the following to the manuscript to
acknowledge this important point.

“The kinetics and steady-state gating behavior predicted by our model were not
quantitatively identical to those in experiments; such discrepancies were expected due
to several overly simplistic assumptions that we used to limit the number of states in
our model. […] Revision of our model to include the influence of PIP2 binding and
the different properties of intermediate- and activated-open states will require
additional studies to better define these properties.”

In response to this interesting point, Seebohm et al. interpret KCNQ1 channel
inactivation using two open states with different conductances that are occupied
sequentially. This idea is consistent with our conceptual model provided the activated
open state of KCNQ1 has a lower conductance or lower open probability than that of the
intermediate open state. However, as we cannot determine which subset of open-states
contribute to the apparent inactivation, we cannot yet use our kinetic model to fit
inactivation. Future studies may provide the necessary data to do so and make this
mechanism clearer.

*2) The single channel properties of KCNQ1 channels differ from KCNQ1/KCNE1
channels. The model should be modified to account for these important differences in
channel properties. One would expect for example two discernible channel open states
for KCNQ1 (corresponding to the intermediate-open (O1) and activated-open (O2)
states), whereas KCNQ1/KCNE1 channels would have only one open state, with properties
similar to the O2 state for KCNQ1 channels. It is stated that “The maximal
current amplitude was increased several fold due to stabilization of the
activated-open state”. Presumably, this means only a change in Po, whereas
experimental data indicate that KCNQ1/KCNE1 currents are larger mainly due to an
increase in single channel conductance (Yang and Sigworth, 1998. J Gen Physiol
112:665)*.

The reviewers are right that we did not clearly address the possible contribution of
changes in single channel conductance to the change in macroscopic current amplitude
when KCNE1 is expressed. Now we acknowledge previous observations that provide
additional explanations for how current amplitude is augmented by KCNE1. (See the
response to the above question.)

We would like to address the statement that a change in single channel conductance has
been already shown to be the main contributor to the difference in KCNQ1 and
KCNQ1+KCNE1 macroscopic current amplitude. The difference in single channel
conductance between KCNQ1 and KCNQ1+KCNE1 was established in two papers using
non-stationary noise analysis (Sesti & Goldstein, 1998; Youshan Yang, 1998). While
these data are enlightening, the assertion that the apparent difference in single
channel conductance quantitatively accounts for all or most of the difference in current
amplitude is uncertain. Current knowledge reveals that there are multiple open states
(current manuscript; Pusch, Ferrera, & Friedrich, 2001; Werry, Eldstrom, Wang,
& Fedida, n.d.) and that there is a fast flickery process in the KCNQ1 pore that is
associated with inactivation and removed by expression of KCNE1 (Pusch et al., 1998;
Seebohm, Sanguinetti, & Pusch, 2003). These data were not available at the time of
the studies using noise analysis, which assumes only a single open state and that the
bandwidth of recording is sufficient to capture all relevant timescales.

Finally, while it may seem simple to define different properties for intermediate-open
and activated-open states, doing so is actually complicated and nontrivial. First, the
single conductance of the intermediate-open and activated-open states is unknown and
exceedingly difficult to measure experimentally. Second, because there are four VSDs,
there are open states with different combinations of VSD states. A fascinating, and
unanswered, question is what properties an open state with a mixed set (for example 2
intermediate and 2 activated) of VSDs will exhibit. Finally, our data indicate that
KCNE1 changes the properties of the O2 (AKA activated-open) state; therefore, it would
not be accurate to assume in the model that the open state of KCNQ1+KCNE1 would
have the same properties as the O2 state of KCNQ1.

*3) The data presented in*
Figure 3
*and corresponding text are incomplete and their interpretation (that it
indicates increased “strength” of VSD-pore coupling in presence of
KCNE1) is not convincing. F351A channels have many properties similar to KCNQ1/KCNE1
channels (especially G-V matching Fmain-V relationship), but the relative Rb/K ion
permeation and pharmacology are not the same. The authors state that F351A like KCNE1
cause significant changes in the Rb/K permeability ratio (*Figure 3*), however this
change is in the opposite direction from KCNE1 (*Figure 2*). This KCNQ1 mutant with
KCNE1-like activation kinetics should have a significantly lower (not a higher)
permeability ratio if it is mimicking KCNQ1/KCNE1 channels. This should be made
clear. Similarly, F351A affects XE991 pharmacology, but this is unlike the effect of
KCNE1. These experiments need to be expanded (e.g., dose-response to XE991;
differences in compound sensitivity requires more than a single concentration) to
more quantitatively describe the effects of KCNE1. In addition, the rubidium data and
pore dilation hypothesis might be better resolved using external cesium, which is
larger, yet permeates homotetrameric KCNQ1 channels*.

1. Increased strength of coupling

The reviewers are right that in the previous version of our manuscript we did not
provide a careful explanation of how the CiVSP experiments substantiate our statement
that KCNE1 strengthens the VSD-pore interaction in the activated-open state. We have now
included additional data to provide evidence of an open-state with high affinity for
PIP2 that is observed only when KCNE1 is coexpressed (Figure 4) and we have revised the text to more clearly explain how we
interpret the data. Paragraph starting, “Does KCNE1 also affect the interaction
between the activated-state of the VSD and the pore to modulate the activated-open
state? […] Altogether these results reveal that KCNE1 causes the activated-open
state to have a high apparent affinity for PIP_2,_ suggesting that KCNE1 may
increase the strength of VSD-pore interactions in the activated open state.”

2. F351A properties are not perfectly matching KCNQ1+KCNE1

This difference was acknowledged in our previous version, although it may not have come
across clearly enough. We do not expect that F351A and KCNE1 would cause exactly the
same changes because F351A causes only a focal change in the VSD-pore interactions while
KCNE1 may cause a more broad change due to its larger interaction surface. We think that
the difference between F351A and KCNE1 actually substantiates a central element of our
manuscript, i.e. that the pore conformation is sensitive to the VSD-pore interactions.
Subjecting these interactions to different perturbations results in different
conformations of the open-pore with various functional consequences. We now discuss
these aspects in two places in the revised manuscript, at the end of the Results and in
the Discussion.

“Of note, these properties of F351A were not identical to those of WT
KCNQ1+KCNE1. However, such differences are not surprising as prior studies(Chan et
al., 2012; Chung et al., 2009; Kang et al., 2008; Lvov, Gage, Berrios, & Kobertz,
2010; Strutz-Seebohm et al., 2011; Y. H. Wang et al., 2011; X. Xu, Jiang, Hsu, Zhang,
& Tseng, 2008; Y. Xu et al., 2013) have located KCNE1 at the VSD-pore interface and
have suggested that KCNE1 engages in a very broad and complex set of interactions with
KCNQ1(Xiaohui Sun, 2012); therefore, it would be unreasonable to expect that a single
point mutation, such as F351A, would alter the VSD-pore interactions in exactly the same
way as KCNE1.”

3. Completion of dose response curve

We thank the reviewers for this suggestion and have added the data as requested (Figure 2, Figure 2—figure supplement 1, Figure 3).

4. Cesium experiment and diameter of the permeation pathway

It is not our understanding that the change in relative permeatility toward Rb, K, or
perhaps other ions, simply reflects a change in the diameter of the permeation pathway.
Here we used the well-studied relative Rb/K permeabilities to probe for different open
states associated with the intermediate and activated conformations of the VSD. The
Cesium study, suggested by the reviewer, may provide additional mechanistic insights of
how ion selectivity is changed amongst different open states in a future study. However,
we believe that it is beyond the scope of this present manuscript.

*4) Several figures use whole cell current amplitude as a proxy for channel
gating. Authors should address the possibility of surface expression level
differences with different proteins, KCNQ mutations and plus/minus KCNE*.

We thank the reviewers for this excellent and important point. We have now included data
(Figure 1—figure supplement 5) to
estimate the surface expression of the different KCNQ1 mutations +/- KCNE1 and show
that the surface expression is similar for the different mutant and is not significantly
altered by KCNE1. Therefore, the differences in current amplitude do not originate from
changes in surface expression, and more likely reflect changes in the gating or
permeation.

*5) CiVSP experiments need to be better explained and analyzed. The protocol for
CiVSP activation is not explained. There is no analysis or evidence of repeated
trials, and the logic is confusing. In the text the claim is made that*
Figure 3
*demonstrates that KCNE1 increases the open state affinity for PIP2. This claim
is not well backed up. Explaining why a lack of effect of CiVSP demonstrates high
affinity rather than no interaction with PIP2 could be helpful*.

We have revised this section to clarify our logic, and we have included additional data
to support the existence of an activated-open state, observed with KCNE1, that has such
high affinity for PIP2 and that PIP2 unbinding is minimal until the channels are closed
(Figure 4). We have also revised the
figures to demonstrate the reproducibility of the response to CiVSP with repeated trials
(Figure 4).

*6) The modeling could be explained better. The modeled effect of KCNQ invokes a
10 fold stabilization of IC and AO states with no alteration of transition barrier
heights. In effect, this results in decreased coupling to the IO state (because the
RO is rarely occupied). The implied change in coupling could be more thoroughly
discussed, as this is the major intellectual contribution of this study*.

We thank the reviewers for their comments. We have expanded the discussion of the model
to illustrate its origins, in which way it differs from previous models, and how a
change in the VSD-pore interactions leads to an apparent change in coupling. We would
also like to mention that, although some previous work has conceptualized coupling as a
barrier between activated and open states in a linear gating scheme, our model describes
coupling as net energy of VSD-pore interactions at different states of these two
structural domains.

“In the voltage-gated ion channel field, VSD-pore coupling, aka electromechanical
coupling, has been a loosely defined term referring to the experimental observation that
pore opening is more likely when the VSDs are activated at depolarized voltages. In our
model of voltage-dependent gating, coupling is represented in a very different way than
in the previously established models. […] It is important to understand that
coupling is a result of all of these state-dependent interactions and a perturbation of
any of these interactions may alter the coupling. These points are illustrated by our
modeling of the KCNE1 effect on KCNQ1 channel gating where strengthening the
intermediate-closed interactions caused an apparent decoupling of pore-opening from the
resting-to-intermediate transition of the VSD, i.e. the open probability was no longer
increased by the transition of the VSD to the intermediate-state at intermediate
voltages. Alternatively, weakening the intermediate-open state interactions would also
decouple opening from the resting-to-intermediate state of the VSD; however, this would
not reproduce the leftward shift of the first FV component that we observed when KCNE1
was expressed (Figure 5).”

*7) While it is mentioned that the kinetics of the model don't entirely
match the data, it appears that a major feature of KCNQ gating is missed by the
model. The “tau slow” of KCNQ (quantified and made note of in the
discussion surrounding*
Figure 1*) is proposed to
be the transition from an IO state to an AO state. The existence of this slow
component is a pillar of their mechanism involving multiple classes of open states.
This slow component seems to be missing from the KCNQ1 modeled kinetics*.

We thank the reviewers for their thoughtful review. The second transition is indeed
present, but it is occurring too quickly to be resolved in the simulated kinetics of
KCNQ1. However, it is readily apparent in the steady-state behavior of KCNQ1, and it can
be more readily appreciated by altering the parameter values to slow the
intermediate-to-activated transition. However, because the parameter space is so vast,
we have not yet identified a solution that provides as clear of a demonstration of the
steady-state behavior when we make such changes. We have not pursued such a solution
further for several reasons. Fist, as described already, our model makes several
simplistic assumptions (PIP2 saturated, all open states have equal conductance, no
inactivation) that make a perfect fit of our model to the experimental data unlikely.
Additionally, we assumed that KCNE1 only increases the intermediate-closed and
activated-open interaction because we have experimental evidence that leads us to that
such changes occur and are critical to the mechanism of KCNE1. However, it is possible
that KCNE1 also affects VSD-pore interactions in other states and those interactions
contribute to gating behavior. Finally, it is important to acknowledge that, because our
model represents the elementary interactions underlying coupling, it is too
underdetermined to identify a single “correct” solution of parameter
values that may be expected to represent the “real” solution. As we do not
intend for our model to be used as a robust gating model that can be plugged into a cell
or tissue model, such a solution may be irrelevant in any case. The true power and
utility of our model is to provide the conceptual framework to explore how
state-dependent interactions contribute to channel gating. For this purpose, we believe
that it does a remarkable job and hope that with additional work in future studies we
can continue to refine the model so that the kinetics, inactivation, and PIP2 regulation
can be better recapitulated by such simple changes in VSD-pore interactions.

*8) Statistics. Students T test was used without explanation of why such a
parametric test is appropriate*.

The Methods section now better explains which tests were chosen for each data set.

“All averaged data reflects n=6 or more from at least two batches of
oocytes. Pairwise comparisons were achieved using Students T test, multiple comparisons
were performed using an Anova with Tukeys Post-Hoc Test. All error bars represent
standard error mean.”

*9)*
Figure 1—figure supplement 4*. Analyzing normalized GVs here would be more informative
than just the IVs in panels (c), (d)*.

The purpose of this figure is to compare the current level of different constructs and
to demonstrate the ohmic nature of E1R/R2E and E1R/R4E with and without KCNE1.
Therefore, we believe that IV is the appropriate format and GV is impossible to
construct.

*10)*
Figure 2*: Explain
voltage protocols used more completely. If KCNQ is slowly transiting from an IO state
to an AO state wouldn't the Rb/K permeability ratio be time and voltage
dependent?*

The reviewers are right that Rb/K permeability should be time and voltage-dependence for
KCNQ1; however, the extent of such effects will depend both on the relative occupancies
and conductances of the different states. Along these lines, previous studies have
described time dependent changes in the sensitivity to intracellular Na ions(Pusch et
al., 2001), which may be related to changes in Rb and K permeation as well. In this
initial description, we used a single voltage protocol to demonstrate the differences in
the conductances of the different states. We now explain this protocol more clearly in
the text.

“Currents from a single cell in external solutions containing 100 mM of
Na^+^, K^+^, or Rb^+^. The currents were
elicited by first stepping the voltage to +60 mV for 5 s then to -60 mV for 3
seconds tails. (b) Averaged Rb^+^/K^+^ permeability ratios
calculated by comparing the tail current amplitudes.”

*11) The authors should discuss/hypothesize how two KCNE1 subunits fit into their
modeling (*Figure 4*) given the recent paper by the Goldstein lab unequivocally
showing that the stoichiometry of the complex is 4:2 Q1:E1*.

Discussion of controversy over stoichiometry and how it relates to our modeling has been
added to the discussion section.

“Central to understanding the modulation of KCNQ1 by KCNE1 is a longstanding
controversy regarding what stoichiometries of KCNQ1:KCNE1 may exist in the fully
assembled channel. Several groups have argued that association of KCNE1 with KCNQ1
dimers during an early stage of biogenesis leads to a fixed 4:2 KCNQ1:KCNE1
stoichiometry and breaks the fourfold symmetry of the channel(Chen, Kim, Rajan, Xu,
& Goldstein, 2003; Morin & Kobertz, 2008; Plant, Xiong, Dai, & Goldstein,
n.d.; K. W. Wang & Goldstein, 1995). […] Fortunately, the F351A mutation
demonstrates that the major finding of our study – the ability to couple pore
opening to different transitions within the VSD – is intrinsic to the KCNQ1
subunit and observed even in the absence of the KCNE1 subunit.”

*12) XE991 Pharmacology: The Wang et al. paper showed that KCNQ1/KCNE1 channels
are 10-fold less sensitive than KCNQ1 channels; however, in*
Figure 2
*wild type KCNQ1/KCNE1 channels do not appear to be 10-fold less sensitive to the
drug. Of all the currents shown, these are the smallest, which makes us wonder if
these are endogenous xKCNQ1 channels assembling with exogenously expressed KCNE1
subunits. Besides the inconsistency with wild type, the E1R/R4E+KCNE1 mutant
should have similar “inhibition” to E1R/R4E. It appears that the
E1R/R4E mutant is modestly activated by XE991*.

We have added additional data (Figure 2—figure supplement 1) and discussion demonstrating that, similar to the report by Wang
et al., we observed that KCNQ1/KCNE1 channels are less sensitive than KCNQ1 alone when
short pulse durations are considered. These differences become insignificant at longer
pulse durations, which is why we did not show a difference in the original version of
the manuscript. As described, we did not expect KCNQ1+KCNE1 to have similar
properties as E1R/R4E because KCNE1 alters the activated open state. However, based on
our hypothesis that KCNQ1+KCNE1 currents come from channels in the activated-open
state, we did expect KCNQ1+KCNE1 to have similar properties as E1R/R4E+KCNE1,
which is what we observed (Figure 2).